# Dynamic and Ballistic Performance of Graphene Oxide Functionalized Curaua Fiber-Reinforced Epoxy Nanocomposites

**DOI:** 10.3390/polym14091859

**Published:** 2022-05-01

**Authors:** Ulisses Oliveira Costa, Lucio Fabio Cassiano Nascimento, Wendell Bruno Almeida Bezerra, Pamela Pinto Neves, Noemi Raquel Checca Huaman, Sergio Neves Monteiro, Wagner Anacleto Pinheiro

**Affiliations:** 1Military Institute of Engineering—IME, Rio de Janeiro 22290-270, Brazil; lucio@ime.eb.br (L.F.C.N.); wendellbez@gmail.com (W.B.A.B.); pamelapneves@gmail.com (P.P.N.); snevesmonteiro@gmail.com (S.N.M.); anacleto@ime.eb.br (W.A.P.); 2Brazilian Center for Physics Research, CBPF, Dr. Xavier Sigaud, 150, Urca, Rio de Janeiro 22290-180, Brazil; noemiraquelchecca@gmail.com

**Keywords:** curaua fiber, epoxy matrix, graphene oxide functionalization, nanocomposite, Izod impact, ballistic performed

## Abstract

Graphene oxide (GO) functionalized curaua fiber (CF) has been shown to improve the mechanical properties and ballistic performance of epoxy matrix (EM) nanocomposites with 30 vol% fiber. However, the possibility of further improvement in the property and performance of nanocomposites with a greater percentage of GO functionalized CF is still a challenging endeavor. In the present work, a novel epoxy composite reinforced with 40 vol% CF coated with 0.1 wt% GO (40GOCF/EM), was subjected to Izod and ballistic impact tests as well as corresponding fractographic analysis in comparison with a GO-free composite (40CF/EM). One important achievement of this work was to determine the characteristics of the GO by means of FE-SEM and TEM. A zeta potential of −21.46 mV disclosed a relatively low stability of the applied GO, which was attributed to more multilayered structures rather than mono- or few-layer flakes. FE-SEM images revealed GO deposition, with thickness around 30 nm, onto the CF. Izod impact-absorbed energy of 813 J/m for the 40GOCF/EM was not only higher than that of 620 J/m for the 40CF/EM but also higher than other values reported for fiber composites in the literature. The GO-functionalized nanocomposite was more optimized for ballistic application against a 7.62 mm projectile, with a lower depth of penetration (24.80 mm) as compared with the 30 vol% GO-functionalized CF/epoxy nanocomposite previously reported (27.43 mm). Fractographic analysis identified five main events in the ballistic-tested 40GOCF/EM composed of multilayered armor: CF rupture, epoxy matrix rupture, CF/matrix delamination, CF fibril split, and capture of ceramic fragments by the CF. Microcracks were associated with the morphological aspects of the CF surface. A brief cost-effective analysis confirmed that 40GOCF/EM may be one of the most promising materials for personal multilayered ballistic armor.

## 1. Introduction

The search for novel materials with optimized properties for applications as ballistic armor has continuously increased due to the corresponding sophistication and power-efficiency of firearms [1,2]. Researchers have studied a wide variety of materials for personal ballistic application. In particular, synthetic polymers such as aramid fiber and ultra-high molecular weight polyethylene (UHMWPE) have been investigated worldwide, and used by police and combat forces, owing to superior mechanical properties combined with lower density as compared to metals and ceramics [3,4,5,6,7].

In the past decade, multilayered armor systems (MAS), comprising a front ceramic followed by a polymer composite reinforced with natural lignocellulosic fiber (NFL) or its fabric, have shown comparable ballistic performance to synthetic polymers such as single plates of Kevlar™ fabric made of aramid fibers [8,9,10,11,12,13,14,15], and Dyneema™, a commercial UHMWPE [16]. As a MAS ballistic second layer, NFL composites display relevant advantages. Indeed, they are environmentally sustainable as well as cost-effective and easy to manufacture [17,18,19,20,21,22]. However, there are disadvantages concerning the surface morphology and the compatibility of natural fibers with polymer matrices; these might impair the integrity of the ballistic target material. For this reason, many superficial treatments were developed to attenuate some of the ballistic effects and prevent loss of target integrity: these treatments can be chemical, mechanical, physical, or biological [23,24,25,26]. All of them are based on removing some of the hydrophilic functional groups present on the surface of the fibers [27]. Improved interfacial properties are crucial to achieving efficient load transmission from matrix to reinforcement, which contributes to reduced stress concentration, and improves the overall mechanical properties. Nevertheless, the influence of all these treatments on mechanical, physical, and dynamic performance has not yet been efficient enough to justify the additional cost.

In recent studies, a possibility of new surface treatment has emerged, and researchers have evaluated the properties of NFLs after functionalization with graphene oxide (GO) [25]. Sarker, et al. [25] indicated that graphene derivatives stand out above all coatings for application in polymer composites. The functional groups present in the GO structure supply the amphiphilicity that enhances the strength of the interface between the polymeric matrix and the natural fibers, and hence optimizes their properties including ballistic performance [28,29,30,31,32,33,34].

Among the NFLs, the curaua fiber (CF) is a promising candidate for replacing synthetic fibers in ballistic composites [35,36,37]. The CFs are extracted from the leaves of the plant *Ananas erectifolius*, which is native to the Amazon region in Brazil. It has been reported that CFs present a density of 0.92 g/cm³, with an average diameter of 50 µm as well as the tensile strength of 1250 to 3000 MPa and Young’s modulus of 27 to 80 GPa [38]. Regarding the GO-functionalized CFs (GOCF), a few studies were recently conducted by Costa, et al. [38,39,40], to evaluate mechanical, thermal, and dynamic mechanical behavior.

As for dynamic mechanical behavior [39], the authors reported significant improvement, of 250% and 90%, in the storage modulus, and of 600% and 200% in the loss modulus, for a 50 vol% CF functionalized with GO reinforcing epoxy matrix composite (50GOCF/EM), in comparison to neat epoxy and 20GOCF/EM, respectively. These results emphasized a significant improvement achieved by GO functionalization in association with the enhanced content of CF. Regarding mechanical properties [40] and ballistic performance [41], which is the focus of the present work, the authors limited their results to epoxy composites reinforced with only 30 vol% of GOCF.

Based on the aforementioned results, the present work studied the absorbed impact energy and the ballistic performance of novel epoxy matrix composites reinforced with 40 vol% CF, both neat and functionalized with GO. In addition to Izod impact tests and high-velocity projectile penetration in the composites, as the MAS second layer, the zeta potential of GO-soaked solution, as well as transmission electron microscopy, field emission-scanning electron microscopy, and electron energy loss spectroscopy, were for the first time used to analyze the GO contribution to the ballistic performance of a nanocomposite. A cost-effective analysis was also performed to reveal the economical advantage of the novel nanocomposite.

## 2. Materials and Methods

### 2.1. Preparation of CF for Functionalization with GO

Curaua fibers (CFs) were supplied by the Federal University of Pará. Initially, the fibers were water-cleaned, then cut to 150 mm and dried in an oven at 80 °C for 24 h. After this process, the CF without treatment were obtained. These fibers were then immersed in a GO solution produced by the Hummers and Offeman method and modified by Rourke, et al. [42], at a concentration of 0.56 mg/mL, remaining under agitation for 1 h in a universal mechanical stirrer, to guarantee and optimize the contact of the GO with the fibers. Then the fibers soaked in this solution were placed in an oven at 80 °C for 24 h, finally obtaining the CF treated with GO (GOCF), corresponding to about 0.1 wt% of GO coating the fibers.

### 2.2. Preparation of Epoxy-Curaua Fiber Composites

The polymer used as the matrix material of the composite plate was the commercial epoxy resin of the bisphenol A diglycidyl ether type (DGEBA), hardened with triethylenetetramine (TETA), using the stoichiometric ratio of 13 parts of hardener to 100 parts of resin. The resin manufacturer was Dow Chemical, Brazil, and the resin was supplied by the distributor Epoxyfiber Ltd.a, Rio de Janeiro, Brazil.

A metal mold with dimensions of 15 × 12 × 1.19 cm was used to manufacture the composites. The molded plates were produced in a 30-ton hydraulic press, using a load of 5 tons for 24 h at ambient temperature. A density of 0.92 g/cm³ was used for the CF as an initial reference [38], while a value of 1.11 g/cm³ provided by the supplier was considered for the epoxy resin (DGEBA-TETA). The percentage of fibers used in this work is 40 vol%, indicated in the nomenclature of 40CF/EM and 40GOCF/EM: the former a composite, while the latter herein referent as a nanocomposite due to the nanometric thickness of GO coating.

### 2.3. Izod Impact Test

Five specimens of both 40CF/EM and 40GOCF/EM conditions were machined according to ASTM D256-10 [43]. Impact tests were performed using a 22 J pendulum Pantec XC-50, Rio de Janeiro, Brazil.

### 2.4. Multilayer Armor Sistem Assembly

An Al_2_O_3_ ceramic doped with Nb_2_O_5_ was used for the manufacture of the first layer of the MAS. Moreover, panels of laminated aramid fabric were used to simulate level IIIA vests with a total thickness of 25 mm.

In the present study, the MAS consisted of 2 layers: a first ceramic layer of Al_2_O_3_ doped with Nb_2_O_5_, and a second composite layer. The layers were fixed together using a polyurethane (PU) glue.

The third layer of aramid fabric was designed to simulate a level IIIA vest according to N.I.J. standard [44]. Thus, a new approach using a ceramics/composite/aramid panel system was set up instead of the ceramics/composite/aluminum alloy system used in previous works [35,36,37,38,39,40,41], as shown in Appendix A.

### 2.5. Ballistic Tests

The ballistic tests were performed on both the 40CF/EM and the 40GOFC/EM composites as MAS second layer, to evaluate the capacity of dissipating kinetic energy and the integrity of the corresponding target. The MAS was placed over a 50 mm thick clay witness (CORFIX™), with similar consistency to the human body. The objective was to obtain the measurement of the backface signature (indentation) caused by the penetration of high-speed (~850 m/s) 7.62 mm caliber ammunition on the witness clay after impact on the MAS target. According to the N.I.J. standard 0101.04 [44], ballistic armor will be effective if the indentation on the clay witness is equal to or less than 44 mm. The measurements were performed with a Q4X Banner digital laser sensor. The tests were carried out at the Brazilian Army Assessment Center (CAEx), Rio de Janeiro.

### 2.6. Zeta Potential Analysis

Zeta potential is a physical property exhibited by any material in dispersion, and is an important parameter used for characterizing the electrical properties of interfacial layers in dispersion. Moreover, it is an important indicator to determine the surface charge of the GO sheets and to characterize the stability of the dispersion [45,46,47]. 

The stability analysis was complemented using characterization by zeta potential from its electrophoretic mobility (Zeta plus—Zeta potential analyzer, Brookhaven Instrument Corporation, Holtsville, NY, USA). The results were reported as an average of ten different measurements and their standard deviation. The zeta potential of all the samples was measured in an aqueous medium according to the ASTM standard [48].

### 2.7. Transmission Electron Microscopy 

For TEM analysis a drop of GO was deposited on ultrathin carbon film with Lacey support. The JEOL 2100F microscope was operated at an accelerating voltage of 200 kV, and equipped with a CMOS camera used in both modes: (i) transmission; and (ii) selected area electron diffraction (SAED). The TEM microscope was also equipped with accessories for energy dispersive X-ray spectroscopy (EDS) and electron energy loss spectroscopy (EELS) (EELS-GIF Tridiem GATAN). The elemental compositions were investigated by EDS and EELS to evaluate the atomic composition using the transmission mode. EELS were conducted using an aperture of 5 mm of the spectrometer. The energy resolution measured by the FWHM of the zero loss peaks was approximately 1.8 eV.

### 2.8. Field Emission-Scanning Electron Microscopy (FE-SEM) Analysis

High-resolution microscopy analysis of the GOCF, with special attention to the GO coating film, as well as of the composite’s ballistic fractured surface, was performed by FE-SEM in a model Quanta FEG 250 FEI microscope operating with secondary electrons accelerated at 30 KV.

### 2.9. Electron Energy-Loss Spectroscopy

Electron energy loss spectroscopy (EELS) is a characterization technique available in the JEOL 2100F TEM equipment. This technique can be applied to obtain essential chemical information, such as the electronic structure of atoms, the covalent bonding, and the nearest-neighbor chemical environment [49]. In one of its many applications, EELS has been used to measure sp2 hybridizations from carbon structures [50]. For carbon-based materials, such as graphene and its derivatives, the electron beam produces two characteristic transitions from internal energy level 1s to unoccupied higher energy states, σ* and π* [51]. These excited states correspond, respectively, to single and double bonds between carbon atoms, and both relate to the excitation fingerprint of the valence band electrons, above the Fermi levels [52]. Consequently, it is possible to know the conversion degree of C-C bonds to C-H bonds by quantifying the sp2 hybridization. One such example of this phenomenon is the oxidation and functionalization processes of natural graphite to obtain GO [53].

## 3. Results and Discussions

### 3.1. Izod Impact Test

Table 1 shows the average Izod impact absorption capacities for the present developed composites 40CF/EM and 40GOCF/EM, in comparison to other reported composites in the literature [52,53,54,55,56,57,58,59,60,61,62,63].

To our knowledge, the average results in Table 1 are superior to any other reported so far for NFL reinforced polymer composites, whatever amount of fiber used, including 40 vol% [54,55,56,57,58,59,60,61,62,63]. Moreover, the only result with NFL functionalized with GO (30 vol% ramie/epoxy) [64] also showed lower Izod impact-absorbed energy. It is noteworthy that even 40 vol% synthetic GO functionalized glass fiber reinforced epoxy composite [65] displays lower Izod epoxy impact-absorbed energy than the present results in Table 1.

Analysis of variance (ANOVA) was used to obtain a more reliable interpretation of the results presented in Table 1, and to confirm if there was a significant difference between the Izod impact-absorbed energy results. According to the calculated ANOVA parameters, F_calculate_ = 8.29 > F_critical_ = 5.32_._ As such, there was, with 95% confidence, a significant difference between the values of 40CF/EM and 40GOCF/EM. In comparison with composites presented in Table 1, nanocomposite 40GOCF/EM can be considered the best so far developed to withstand dynamic impact loading.

### 3.2. Ballistic Impact Results

Figure 1 shows the appearance of the MAS target after being hit by a 7.62 mm projectile in the ballistic tests using 40CF/EM and 40GOCF/EM as second layer. In both Figure 1a,b it is noteworthy that the front ceramic tile, illustrated in Appendix A, has completely disappeared. However, the composite plate maintained its integrity, except for the hole at its center associated with the projectile’s complete penetration of the target.

Table 2 presents the ballistic backface signature, i.e., the depth of penetration in the witness clay backing the MAS target, caused by the 7.62 mm projectile impact, as described in Section 2.5.

Figure 2 shows the graphical representation of the results in Table 3 compared to the criterion (44 mm) established by the NIJ standard [44] for the maximum allowed penetration without lethal trauma (red dashed line).

The ANOVA for the backface results for both 40CF/EM and 40GOCF/EM in Table 2 and Figure 2 indicated, with a confidence level of 95%, that the values were equal, since F_calculated_ = 1.57 < F_critic_ = 2.62. The similar results for the backface signature of both 40CF/EM and 40GOCF/EM were not surprising since not only the composite plate affects the projectile penetration but also the front ceramic and the aramid fabric used as a third layer in the armor vest. The important point is that the values for our novel composites are comparable with most MAS with NFLs composites as a second layer.

Table 2 also lists backface signatures of distinct NFLs composites, as well as Kevlar™ plate as the MAS second layer, all with same thickness of 10 mm, as well as that of 25 mm plate Dyneema™ replacing a complete MAS.

By comparing the present results in Table 2, it is clear that increasing the amount of fiber to 40 vol% did not reduce the ballistic performance, which is an advantage in terms of cost-effectiveness, as further discussed. Indeed, the backface signature of our novel composites, 24.8 mm, as MAS second layer against high-speed 7.62 mm rifle bullet, was significantly below the 44 mm limit for lethal trauma. This is much better, for instance, than single plate of Dyneema™ used worldwide for this kind of protection.

### 3.3. Cost-Effectiveness Analysis

A brief weight and cost analysis for the corresponding MAS, Table 3, was found to favor the 40 vol% curaua fiber epoxy composite. Densities were obtained from experimental and theoretical measurements, while prices were based on January 2022 commercial values. Although the front ceramic used in the ballistic test was a small hexagonal plate, its calculated face area was considered to cover the whole 15 × 15 = 225 cm^2^ surface of the armor, which would correspond to a real situation. The bottom of Table 3 reveals that the substitution of CF composite, as the MAS second layer, for an equal thickness of conventional Kevlar™ caused a slight decrease in weight of about 0.81% and a decrease in cost of 1.24%. These results indicate that, in practice, a MAS with 40 vol% CF composite, instead of 30 vol% CF composite, provides optimized ballistic performance, as well as lightness at a lower cost. The 40GOCF/EM composite showed enhanced properties in terms of impact performance in comparison to the 40CF/EM composites. As can be seen from Table 3, there was no change in weight between the 40GOCF/EM and the 40CF/EM composites. However, there was a slight increase in cost due to the functionalization with GO, the price of which depends on the market. However, the amount used, ~0.1 wt%, had a negligible modification on the composite price. Nevertheless, the addition of GO was crucial to improve the dynamic impact properties.

The GO functionalization of NFLs reinforcing polymer matrix composites is now well-established as an effective way to improve properties [28,29,30,31,32,33]. The present work provides evidence that this is the case with novel 40 vol% CF reinforced composites, regarding dynamic impact resistance. Although a relatively small amount, 0.1 wt%, of GO is incorporated into the composite, more detail is needed to understand its possible contribution to the composite performance.

### 3.4. The Zeta Potential of GO Dispersion

The zeta potential of 10 samples of the GO dispersion presented a negative value of −21.46 ± 2.93 mV, which was assigned to the ionization of electronegative functional (OH and COOH) groups formed at the graphite lattice during the oxidation [45,46,47]. 

Moreover, this result was slightly lower than those found in the literature for GO with pH around 4 and the degree of oxidation in accordance with Krishnamoorthy, et al. [45,46,47]. Regarding the stability of the colloidal suspension, and considering ASTM Standard D 4187–82 [48], a zeta potential between 30 and −40 mV (either positive or negative) showed moderate stability; higher than 40 mV (either positive or negative) resembled high stability [4]. In order to compare values, reduced graphene oxide (RGO) dispersions usually present lower mean values of Zeta potential of about −3.81 mV [46]. In the present study, the GO dispersion presented relatively low stability. This could be explained by the presence of more multilayered structures than monolayer or few-layer flakes of GO, which contributed to agglomeration, as could be later seen in the SEM analysis.

### 3.5. Electron Microscopy Analysis of GO

In Figure 3a, a low-magnification SEM image, the GO appears as multi-layered stacked flakes. This corroborates the zeta potential analysis, Section 3.4, showing that the dispersion is not very stable due to the size and thickness of the flakes. As can be seen in Figure 3b, in a low-magnification TEM image, the GO is highly transparent to electrons, even compared to thin-film carbon support. However, a contrast between areas 1 and 2 can be observed due to the number of layers of the GO film. The darker region indicated as area 1 has a higher number of layers compared to area 2. This corroborates the EDS result shown in Figure 3c, which reveals that area 1 has much more carbon and oxygen than area 2. An SAED pattern of both areas of the GO film is shown in Figure 3d,e. Surprisingly, clear diffraction spots are observed as characteristic of some crystalline order; the six-fold pattern is consistent with a hexagonal lattice, and the points are labeled according to Miller–Bravais hkl notation, such as direction [001] in Figure 3d and plains, (100) and (110), in Figure 3e. Regarding area 2, in Figure 3c the individual spots are now barely visible as the contributing patterns merge into a ring pattern characteristic of a polycrystalline sample. In both areas 1 and 2, samples were observed in a gold-sputtered grid. This suggests that there are no preferred stacking orientations between the GO monolayers when the thin coating film is formed in this way [73].

The diffraction patterns obtained for both areas agreed with the patterns found by Wilson, et al. [73]. The hexagonal pattern of sharp spots was similar to those obtained for graphite oxide, based on two reasons. Firstly, the GO sheets were not completely amorphous. In fact, sharp spots indicate short-range order over a length scale of the coherence length of the electron beam, which is only a few nanometers under the conditions used here. Secondly, long-range orientational order was present, which extended at least as far as the width of the selected area aperture (ca. 0.6 m), since any rotation of the sheet would appear as smearing or splitting of the spots.

The hexagonal pattern of sharp points and ring pattern is similar to that obtained from GO. According to Wilson, et al. [73] and Meyer, et al. [74], the diffraction patterns indicate that the GO film is not totally amorphous, i.e., it presents monocrystalline (area 2) and polycrystalline (area 1) regions, with no preferred orientation, which is characteristic of multilayered GO films. This can be confirmed through the diffraction pattern, because the inner reflections are more intense than the outer ones. Interplanar spacings (d-spacings) can also be extracted from electron diffraction patterns. As can be seen from Figure 3f, the interplanar distances for the GO planes {100} and {110} were 0.235 and 0.142 nm, respectively, and they were similar to the distances found by Wilson, et al. [73].

Figure 4 shows the thickness of GO film (25–30 nm) coated onto the CF surface measured by a FE-SEM with a magnification of 10,000×. This result indicates that the GO film is composed of more than 30 layers [73]. Apparently, the GO functionalization is not totally homogeneous on the fiber surface due to the stacking of the GO flakes in the dispersion. This could be noticed from the FE-SEM image of the GO dispersion, i.e., it is comfirmed that the GO is responsible for linking the curaua fiber to the epoxy matrix at some points. Thus, this anchoring might cause an effective improvement in interfacial strength, and consequently in mechanical and dynamical impact properties.

### 3.6. Electron Energy Loss Spectroscopy (EELS) Analysis of GO

Figure 5a shows the energy loss spectra of Carbon (C) K-edge and Oxigen (O) K-edge probed at different areas (Area 1 and Area 2) of the GO sample. In this figure, it is possible to identify the individual oxygen peak at approximately 540 eV. This agrees with the values reported in the literature [51]. The carbon core-loss spectra probed at the two different areas are shown in Figure 5b. At an energy loss of 284 eV, the 1s to π* transition is observed, while the related 1s to σ* transition states were observed starting at approximately 291 eV, also in accordance with the literature [49,52,75]. When comparing the two spectra in Figure 5b, C K-edge and O K-edge, a noticeable change in intensity can be observed between the probed areas (Area 1 and Area 2). It has been reported that this variation can be attributed to the difference in thickness of the samples [52]. Considering that the intensity of the core edge decreases with increase of the sample thickness, it can be concluded that area 2 is thinner than area 1. A similar difference in intensity has also been observed between single and multilayered GO, with the latter having higher intensity, which may imply that the variance in thickness is associated with the number of layers in the GO structure [52].

### 3.7. Ballistic Fracture Characterization

SEM was also used to characterize the ballistic fracture of the novel composite. As shown in Figure 1, the 40CF/EM and 40GOCF/EM composites, after the 7.62 mm projectile shooting against the MAS target, remained relatively intact, with little delamination, and being able to withstand more than one ballistic event of the same or less impact energy. 

Regarding the hexagonal ceramic layer, Appendix A, it was destroyed with the impact of the projectile. In a real vest, these tiles compose a mosaic to allow multiple shots in which each tile may be hit at a given time without affecting the other tiles and compromising the entire armor integrity.

Figure 6 shows the SEM of the fracture surface of the destroyed MAS front ceramic. This rupture occurred due to intergranular fracture, absorbing most of the projectile’s kinetic energy, similar to that reported by other authors [76,77].

Another relevant aspect of the novel composites plates as MAS second layers was the capture of ceramic fragments resulting from the shattered frontal ceramic, which corresponded to a significant amount of the absorbed impact energy [5].

Figure 7a illustrates the capture of ceramic fragments by the fibrils that composed each CF in the epoxy composite. In this figure the mechanisms of: (i) rupture of fibers; (ii) rupture of the matrix; (iii) separation of fibrils; (iv) delamination, from both the projectile and (v) the ceramic plate, are also observed [41].

In contrast to Kevlar™ as a second layer of a MAS [39,40,41], the CF-reinforced composite has several mechanisms with significant influence of the GO coating. The combination of the energy dissipation mechanisms such as fragment capture, fibril separation, delamination, fiber rupture, and matrix rupture make the 40CF/EM and 40GOCF/EM composites an effective ballistic protection as Kevlar™ [41].

From Figure 7b one can observe that the fiber surface morphology caused stress-concentrating points in the composite matrix, which promoted nucleation and the propagation of microcracks in specific regions. 

## 4. Conclusions

Novel epoxy nanocomposite reinforced with graphene oxide (GO) functionalizing 40 vol% curaua fiber (40GOCF/EM), compared with non-functionalized (GO-free) 40CF/EM composite, was investigated for dynamic impact and ballistic performance. On average, the 40GOCF/EM presented absorbed-impact energy, 813 J/m, superior to any other natural fiber composite, either GO-free or functionalized, reported so far. The ballistic performance of both 40GOCF/EM and 40CF/EM, as second layer in a ceramic front-multilayered armor system (MAS) against 7.62 mm level III ammunition, displayed the same 24.8 mm backface signature, which is comparable with other MAS with natural fiber composites. A major advantage of the novel composites is their cost-effectiveness. The contribution of GO was analyzed by the zeta potential as well as TEM, SEM, and EELS, revealing the presence of several stacked layers rather than few-layered GO flakes. EDS and SAED results suggest no preferred stacking orientation between the GO layers when nanometric films are formed on the curaua fibers. The novel composite plates remained intact after the ballistic test. Combined energy dissipation mechanisms associated with ceramic fragment capture, fibers separation, fiber-matrix delamination, and fiber and matrix rupture were for the first time revealed by SEM. The novel 40GOCF/EM nanocomposite is one of the most promising types of MAS second layer.

## Figures and Tables

**Figure 1 polymers-14-01859-f001:**
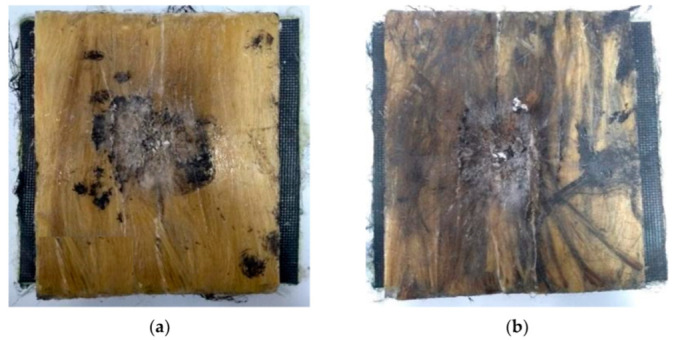
Front view of the MAS target after the ballistic test. The second layer corresponds to composite plates: (**a**) 40CF/EM and (**b**) 40GOCF/EM.

**Figure 2 polymers-14-01859-f002:**
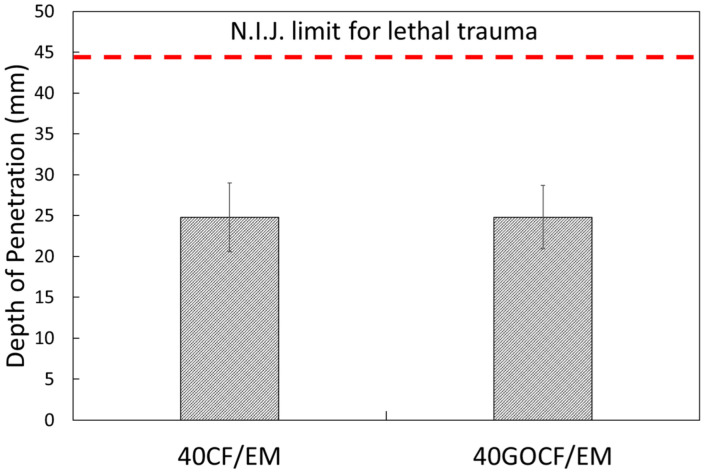
Graphical representation of the depth of penetration of backface signature results.

**Figure 3 polymers-14-01859-f003:**
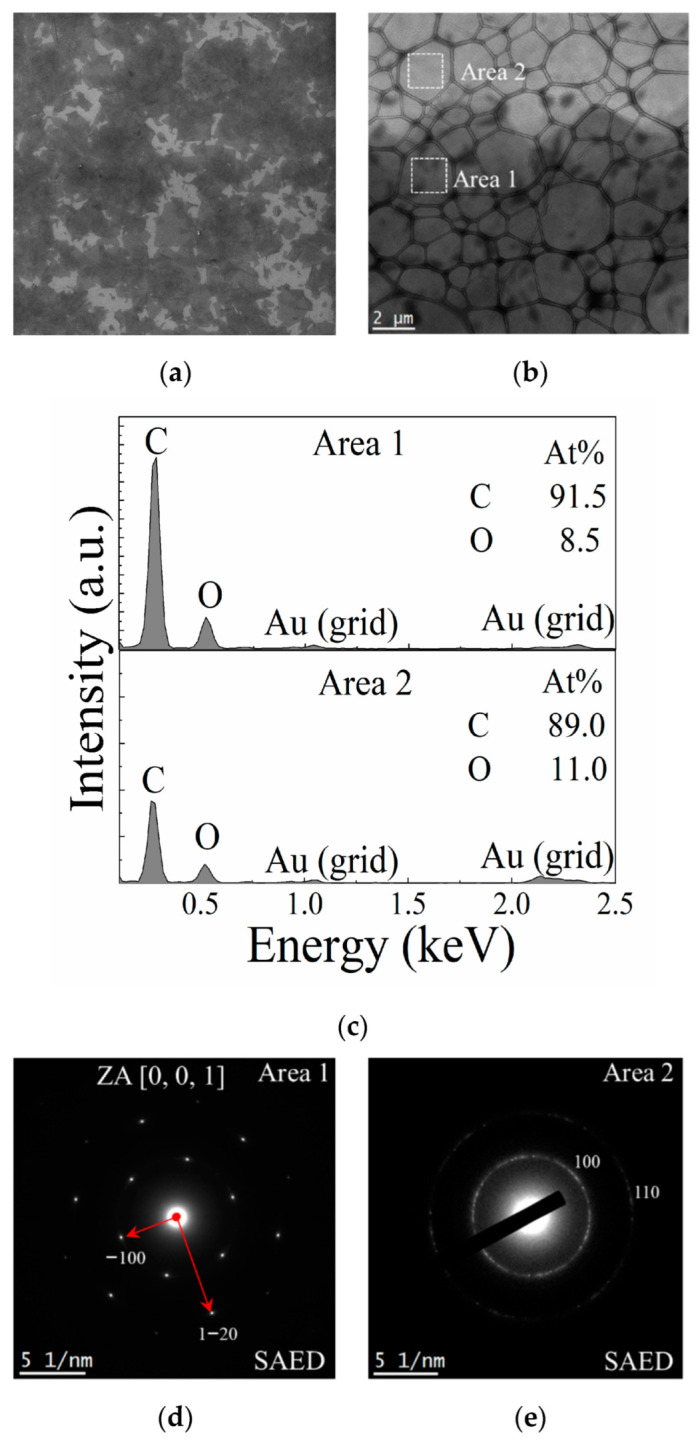
(**a**) SEM image of the dispersion of GO flakes on a silicon substrate; (**b**) TEM image of a GO film with two areas of different hues; (**c**) EDS of both selected areas of the GO film (**d**,**e**) SAED of areas 1 and 2 respectively, the diffraction points are marked with Miller-Bravais indices; (**f**) Intensity profile and interplanar distances of the indexed planes through the diffraction points indicated in the panel of areas 1 and 2.

**Figure 4 polymers-14-01859-f004:**
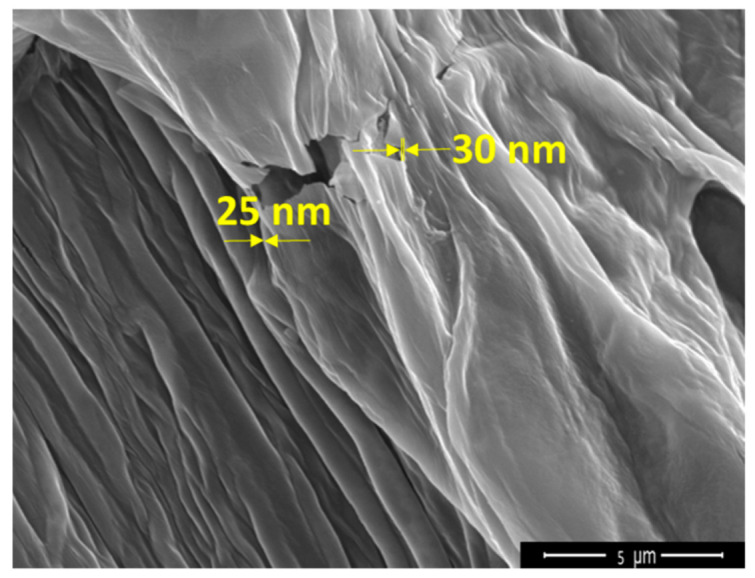
High-magnification FE-SEM image of GO-coated curaua fiber.

**Figure 5 polymers-14-01859-f005:**
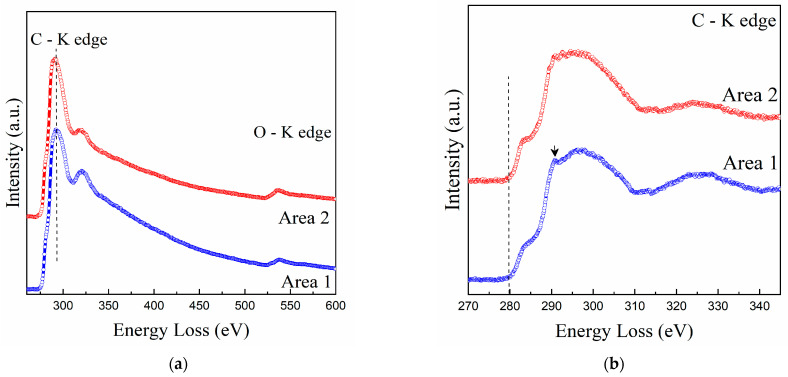
EELS analysis of GO at (**a**) and carbon core-loss spectra (**b**) probed at two different areas, 1 and 2, indicated in Figure 3.

**Figure 6 polymers-14-01859-f006:**
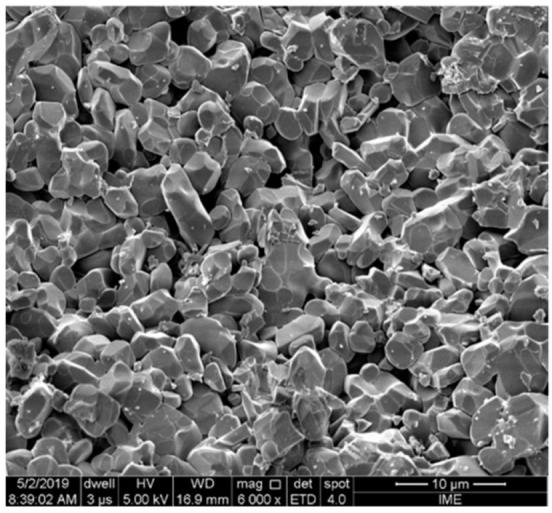
Fracture surface of the MAS front ceramic hexagonal tile.

**Figure 7 polymers-14-01859-f007:**
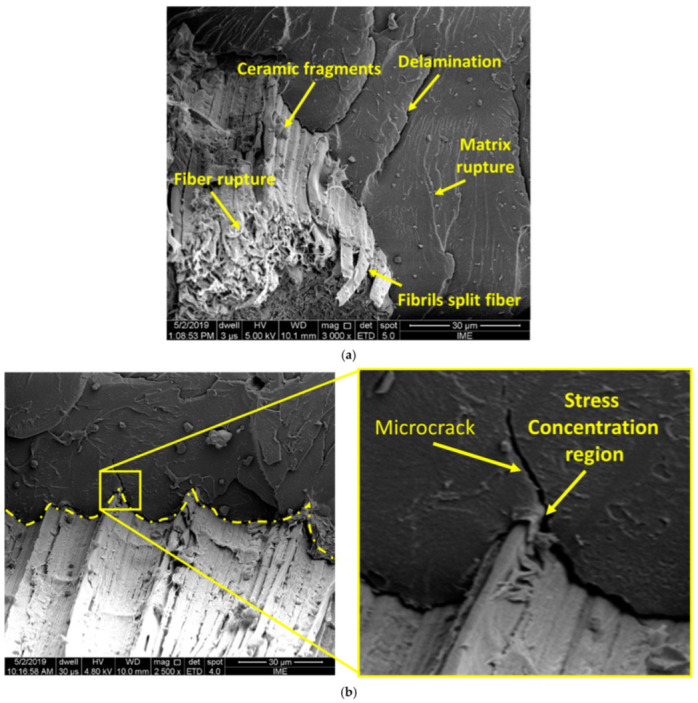
Fracture surface by SEM analysis: (**a**) Fiber rupture, matrix rupture, delamination, capture of the fragments, and fibrils split fiber mechanisms, (**b**) microcrack propagation in the composite matrix.

**Table 1 polymers-14-01859-t001:** Izod impact-absorbed energy for different fiber-reinforced composites.

Fiber/Matrix	Amount of Fiber	Izod Impact- Absorbed Energy (J/m)	Reference
Curaua/Epoxy	40	620	PW
GO-functionalized Curaua/Epoxy	40	813	PW
Ramie/Epoxy	30	567	[52]
Mallow/Epoxy	30	499	[53]
Tucum/Epoxy	40	216	[54]
Sedge/Epoxy	30	63	[55]
Fique/Epoxy	40	222	[56]
Ramie/Polyester	30	594	[57]
Curaua non-woven/Epoxy	30	433	[58]
Banana Fiber/Polyester	30	265	[59]
Curaua/Polyester	30	190	[60]
Jute/Epoxy	30	426	[61]
GO Ramie/Epoxy	30	60	[62]
GO Glass Fiber/Epoxy	40	290	[63]

PW—Present Work.

**Table 2 polymers-14-01859-t002:** Backface signature of NFLs composites and Kevlar™, all in 10 mm thick plate, composing the MAS second layer, as well as 25 mm thick Dyneema™ plate, completely replacing the MAS target. Plates ballistic tested against 7.62 mm N.I.J [44] level III ammunition.

Material in Ballistic Armor for Protection against Level III Ammunition	Backface Signature (mm)	Reference
40 vol% curaua fiber/epoxy composite	24.8	PW
40 vol% GO-functionalized curaua fiber/epoxy composite	24.8	PW
30 vol% Guaruman fiber/epoxy composite	27.5	[17]
30 vol% pineapple leaf fiber (PALF)/epoxy composite	26.6	[16]
30 vol% curaua fiber/epoxy composite	24.3	[35]
30 vol% curaua fiber/polyester composite	22.2	[66]
30 vol% jute mat/polyester composite	24.7	[67]
30 vol% coir fiber/epoxy composite	31.6	[68]
30 vol% sisal fiber/ Polyester composite	22.3	[69]
30 vol% bamboo fiber/epoxy composite	18.2	[70]
30 vol% sugarcane bagasse/epoxy composite	39.8	[71]
10 mm Kevlar™ as the MAS second layer	21.3	[72]
25 mm Dyneema™ plate (complete MAS)	41.5	[16]

PW—Present Work.

**Table 3 polymers-14-01859-t003:** Weight and cost analysis for a 150 × 150 mm MAS with aramid 30 vol% or 40 vol% curaua fiber epoxy composite.

Armor Component	Volume (cm^3^)	Density (g/cm³)	Weight (kg)	Price per kg (US Dollars)	Component Cost (US Dollars)
Al_2_O_3_-ceramic tile	225	3.89	0.74	2.60	1.94
30CF/EM	225	1.05	0.24	25.91	6.22
40CF/EM	225	1.03	0.23	25.91	5.96
GO	-	-	0.000083	31.240 ^a^–174,000 ^b^	2.59–14.44
40GOCF/EM	225	1.03	0.23	28.50–40.35	6.56–9.28
Kevlar™	168.75	1.09	0.18	72.50	13.05
Total MAS weght with 30CF/EM	1.23		Total cost with 30CF/EM composite (US dollars)		21.21
Total MAS weight with 40CF/EM composite (kg)	1.22		Total cost with 40CF/EM composite (US dollars)		20.95
Total MAS weight with 40GOCF/EM nanocomposite (kg)	1.22		Total cost with 40GOCF/EM composite (US dollars)		21.55–24.27
Decrease in weight (%) 30CF/E compared to 40CF/EM	0.81		Decrease in cost (%) 30CF/E compared to 40CF/EM		1.24
Decrease in weight (%) 40CF/E compared to 40GOCF/EM	0.00		Increase in cost (%) 40CF/E compared to 40GOCF/EM		2.86–15.85

^a^https://www.go-graphene.com (accessed on 19 April 2022), ^b^
https://www.acsmaterial.com (accessed on 19 April 2022).

## Data Availability

Not applicable.

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
