# Peer review of "Dynamic and Ballistic Performance of Graphene Oxide Functionalized Curaua Fiber-Reinforced Epoxy Nanocomposites"

_polymers, 2022, doi:10.3390/polym14091859_

Round 1

Reviewer 1 Report

Costa et. al reported work "Dynamic and Ballistic Performance of Graphene Oxide Functionalized Curaua Fiber-Reinforced Epoxy Composites" is well written can be acceptable upon addressing minor comments.

1) Some of the recent references related to GO-Epoxy functionalised composites for Ballistic performance applications were neglected. It need to be cited wherever it looks appropriate in the introduction. 

2) There are some typographical mistakes, in the manuscript, need to be corrected. 

3) Ballistic performances of GO-Epoxy functionalized materials (previously reported) were need to be tabulated and highlight the importance of current work.

4) Along with SEM, XPS spectra need to be incorporated to characterize the composite materials.  

Author Response

Response to the Reviewers

The authors would like to thank the Reviwers for the valuable comments and suggestions on the structure and scientific aspects that contribute to improve the manuscript. Amendments are provided accordingly. Responses to each comment are listed below and all modifications/additions were marked as Track changes in the revised version of the manuscript.

Reviewer #1

General comment: Costa et. al reported work "Dynamic and Ballistic Performance of Graphene Oxide Functionalized Curaua Fiber-Reinforced Epoxy Composites" is well written can be acceptable upon addressing minor comments.

Response: The authors would like to thank for the compliment made by the Reviewer. Moreover, all modifications were marked as Track changes in the revised version of the manuscript.

Comment (1): Some of the recent references related to GO-Epoxy functionalised composites for Ballistic performance applications were neglected. It need to be cited wherever it looks appropriate in the introduction. 

Response: The Reviewer is right about the Introduction. Hence, a new Introduction is now in the revised version of the manuscript contemplating recent references that are important for our work.

Comment (2):  There are some typographical mistakes, in the manuscript, need to be corrected. 

Response: Complied, a new corrected text of the manuscript is now presented in the revised version.

Comment (3):  Ballistic performances of GO-Epoxy functionalized materials (previously reported) were need to be tabulated and highlight the importance of current work.

Response: The authors agree with the Reviewer and new tables comparing the results obtained in this study with some reported in other recent studies are now in the revised version.

Comment (4): Along with SEM, XPS spectra need to be incorporated to characterize the composite materials.

Response: As recommended, new FE-SEM images and TEM are now in the revised version. Our ongoing work will present XPS spectra to characterize the 40GOCF/EM nanocomposite.

Reviewer 2 Report

Title: Dynamic and Ballistic Performance of Graphene Oxide Functionalized Curaua Fiber-Reinforced Epoxy Composites

The novelty of this work, as well as its interesting results, is clearly described in the article. Therefore, the referee would like to recommend this work to minor revision and to be published after consideration according to the comments below:

  1. Please add more current papers in the literature and improve introduction section. Some interesting papers related to the topic of this manuscript could be:

Experimental and numerical study on HDPE/SWCNT nanocomposite elastic properties considering the processing techniques effect, Microsystem Technologies 26 (8), 2423-2441.

  1. The writing of this paper needs to be improved and polished. Some clumsy and neglectful expressions can be found.

Author Response

Reviewer #2

General comment: The novelty of this work, as well as its interesting results, is clearly described in the article. Therefore, the referee would like to recommend this work to minor revision and to be published after consideration according to the comments below:

Response: The authors agree with the reviewer, and a new version of the manuscript is now provided. In this new version, more experiments were performed including some interesting results about 40 vol% GOCF nanocomposites.

Comment (1): Please add more current papers in the literature and improve introduction section. Some interesting papers related to the topic of this manuscript could be:

Experimental and numerical study on HDPE/SWCNT nanocomposite elastic properties considering the processing techniques effect, Microsystem Technologies 26 (8), 2423-2441.

Response: As recommended, new and current papers are now included in the revised version of the manuscript.

Comment (2): The writing of this paper needs to be improved and polished. Some clumsy and neglectful expressions can be found.

Response: The authors fully agree with the comment made by the reviewer and the new version of the manuscript is now revised, improved, and polished.

Reviewer 3 Report

This manuscript does not have worthy to publish in a polymer journal (Q1 journal). The manuscript doesn't meet the scientific evidence. There is no novelty in this paper. Results are limited, and in general. there is no interesting result. many places have over statements. In summary, the article has serious flaws, additional experiments needed, research not conducted correctly. 

Author Response

Reviewer #3

General comment: Costa et al. fabricated GO, curaua fiber and epoxy composite and study of its dynamic and ballistic performance. This study lacks so many analyses. Manuscript have no sufficient information for readers. Therefore, Manuscript need to carefully revised before publication.

Response: The authors agree that our manuscript lacks some other analyses and needs to improve additional information for the readers. A careful revision was performed in order to attend all points and questions raised by the Reviewer.

Specific comment: Line 98, “a GO solution produced by the Hummers and Offeman method and modified by Rourke et al. (2011)” how authors prepare GO solution. Is it commercial or synthesized in lab? Please provide details of the material. How to modify. Provide optical images of GO. Analysis the GO using XRD TEM and RAMAN spectroscopy at least. After fiber immersed please provide the zeta potential data how fiber is soaked with GO. Provide its mechanism.

Response: Detailed information on GO is now indicated in the methodology with both optical and SEM images as well as additional TEM and EELS spectroscopy results. Finally, the zeta potential and mechanism of fiber functionalization with GO are presented in the revised version.

Comment (2): What is the novelty of this study? Authors similar manuscript, same work already published on Polymers journal and other journal. I don’t find any difference to publish this manuscript in polymers journal. May be only fig. 5 is the difference.

Response: The authors apologize for not emphasizing the novelty of our work as compared to previous ones published in Polymers and JMRT. Now, we make it clear that not only an enhanced content of reinforcement (40 vol% of GO_functionalized curaua fiber) but also different characterization techniques recommended by the Reviewer are demonstrating the relevance of our novel nanocomposite as a promising ballistic armour.

Comment (3): Section 2.2 epoxy-GOCF composite samples need additional experiments (TGA, DSC or DMA for calculation Tg, surface roughness, XPS, FTIR). Identify where is the GO matrix by using FE-SEM. Due to GO and fiber mainly Ballistic was performed.

Response: As recommended, additional experiments were performed in the novel 40 vol% GOCF-epoxy nanocomposite for the calculation of important parameters. The GO is now better identified by FE-SEM, TEM, and EELS as well as its role in the ballistic results. As for TGA, DSC, DMA, XPS, and FTIR our ongoing research work will disclose corresponding results, which are not essential for the present evaluation of 40GOCF/Epoxy performance as ballistic armor.

Comment (4): Line 114, authors written fibers in this used are 20, 30 and 40%, which is so high amount? is it GOCF or CF/E only fiber? totally unclear that information in the manuscript.

Response: The authors fully understand the doubt about the amount of reinforcement (GOCF) in the epoxy matrix. To clarify this point, only the novel 40vol% GOCF nanocomposite is now considered in the revised version. The reason for this relatively high amount and its advantages are discussed.

Comment (5): In Table 1, authors provided 5 specimens each samples, there are huge difference in all specimen which is unacceptable for experiment in lab. Reproducibility of this material is so low. Only present mean ± SD. Other details no need. But SD is so high which is totally unacceptable.

Response: The reviewer is right on the low reproducibility of our composites. New impact tests allow for an improvement of our results, which are now presented only by mean value ± SD.

Comment (6): Section 2.4now well written. Difficult to understand. Please rewrite it again.

Response: As requested, Section 2.4 is rewritten in the revised version for better understanding.

Comment (7): Fig. 1 is general so move to supplementary file

Response: Complied, Fig. 1 is now in the manuscript S1.

Comment (8): Table 2, no need to insert manuscript.

Response: As recommended, Table 2 is now deleted and only its relevant data is presented and discussed.

Comment (9): In table 3, line above 175 row no need to provide.

Response: complied, line above 175 is now deleted.

Comment (10): Where is sample and where is data totally unclear in table 3.

Response: Table 3 is now modified to make clear where are samples and data.

Comment (11): Section 3.1 so many paragraph, two sentence do not make one paragraph.

Response: As suggested, the number of paragraphs in Section 3.1 is now reduced in the revised version.

Comment (12): In table 4 only data SD need no need other information and compare statically which data is significant which data.

Response: As recommended, only mean and corresponding SDs are now shown in Table 4.

Comment (14): Table 5 no need.

Response: As recommended, Table 5 is now deleted and only its relevant data is presented and discussed.

Comment (14): Section 3.2, not necessary so many subsections.

Response: As pointed out by the Reviewer, the authors agree that there are too many subsections in Section 3.2. They are now excluded in the revised version.

Comment (15): In conclusion very less information need to rewritten in one paragraph.

Response: Conclusions are now rewritten in one paragraph, as recommended, and added with more information.

Comment (16): Compare this study with other studies in a tabular form.

Response: This is an excellent suggestion and a new table compares our present work with other related studies.

Comment (17): Introduction need to rewritten briefly with the aim of materials and novelty of this study.

Response: As recommended, the Introduction is now briefly modified with emphasis on the novelty of our work.

Comment (18): Reference style is totally different than journal guidelines. Please double check author guidelines. Check previous paper which are published in Polymers journal

Response: The Reviewer is absolutely right, our reference style was not complying with Polymers guidelines. This has been thoroughly corrected in the revised version.

Round 2

Reviewer 3 Report

The Authors improved the manuscript as compared to the original submission. However, still there are many points that need to correction before final publications.

Comments:

1. Conclusions is too much and many paragraphs again, please make it one paragraph and include a few important results (preferably with 250-200words)

“Comment (15): In conclusion, very less information needs to rewritten in one paragraph. Response: Conclusions are now rewritten in one paragraph, as recommended, and added with more information”. But here authors have written, it is one paragraph which is ridiculous.

  1. Line 69, and other places need references, authors can indicate the following references Cellulose, 29 (2022) 2399-2411 and Journal of Industrial and Engineering Chemistry 21 (2015) 11-25
  2. Make Table 1 and 2 into one table, table 1 your study mention ‘This study’. Similarly, Tables 3 and 4.
  3. Figure clarity is too low, even in pdf files not visible, especially fig. 2, fig 3 (c) and (f), fig 5,
  4. All the subfigure label should be indicated inside the text. Please do it.
  5. Table 6 is useless with only one sample and one information. Please remove it and the information mentioned in the text.
  6. Section 3.4 needs to be rewritten in a better way.
  7. Check carefully for spelling errors and consistency of all abbreviations throughout the manuscript.

Author Response

Comments Reviewer #3

Comment 1: Conclusions is too much and many paragraphs again, please make it one paragraph and include a few important results (preferably with 250-200words)

Response: In the new version of the manuscript the conclusions is now in one paragraph with 202 words.

“Comment (15): In conclusion, very less information needs to rewritten in one paragraph. Response: Conclusions are now rewritten in one paragraph, as recommended, and added with more information”. But here authors have written, it is one paragraph which is ridiculous.

Response: The authors would like to apologize because we understood that the Reviewer requested to format the style of the manuscript conclusions for paragraphs instead of point by point. In the new revised version, this is now was corrected.

Comment 2: Line 69, and other places need references, authors can indicate the following references Cellulose, 29 (2022) 2399-2411 and Journal of Industrial and Engineering Chemistry 21 (2015) 11-25

Response: Complied, these references are now included in the manuscript.

Comment 3: Make Table 1 and 2 into one table, table 1 your study mention ‘This study’. Similarly, Tables 3 and 4.

Response: The Reviewer is totally right and those tables are now combined in this better way in the revised version.

Comment 4: Figure clarity is too low, even in pdf files not visible, especially fig. 2, fig 3 (c) and (f), fig 5.

Response: The authors are sorry for this inconvenience, we sent all figures and tables in on .ZIP file and all figures were in more than 400 dpi as is required for the journal. However, we improve the quality of all figures as requested by the Reviewer.

Comment 5: All the subfigure label should be indicated inside the text. Please do it.

Response: Complied, the new version of the text is now composed with the subfigure labels.

Comment 6: Table 6 is useless with only one sample and one information. Please remove it and the information mentioned in the text.

Response: The authors totally agree with the Reviewer and the table is now deleted from the manuscript as well as its mention in the text.

Comment 7: Section 3.4 needs to be rewritten in a better way.

Response: A new version of the section 3.4 is now presented in the revised version.

Comment 8: Check carefully for spelling errors and consistency of all abbreviations throughout the manuscript

Response: The Reviewer mentioned an interesting last thing to do in order to improve the text. The spelling errors and abbreviations are now checked.

Round 3

Reviewer 3 Report

The revised revision has been improved to a satisfactory level. Now it can publish.